# Effect of Mg Treatment on the Nucleation and Ostwald Growth of Inclusions in Fe-O-Al-Mg Melt

**DOI:** 10.3390/ma13153355

**Published:** 2020-07-28

**Authors:** Yutang Li, Linzhu Wang, Chaoyi Chen, Junqi Li, Xiang Li

**Affiliations:** 1School of Materials and Metallurgy, Guizhou University, Guiyang 550025, China; DomYutangLi@163.com (Y.L.); ccy197715@126.com (C.C.); jqli@gzu.edu.cn (J.L.); 2College of Materials & Metallurgical Engineering, Guizhou Institute of Technology, Guiyang 550003, China; lixiang8656@163.com

**Keywords:** Mg treatment, inclusion, nucleation, growth, holding time (second)

## Abstract

This study aimed to investigate the effect of Mg treatment on the nucleation and ostwald growth of inclusions. Deoxidized experiments with Al (0.05%Al) and Al-Mg (0.05%Al + 0.03%Mg) were carried out at 1873 K, and the composition, number, and size of inclusions were studied as a function of holding time. Homogeneous nucleation theory and ostwald ripening were utilized to calculate the nucleation rate, the critical size of nuclei, and coarsening rate of inclusions. The results show that small inclusions were more easily found in the steels with Al-Mg complex deoxidation, and the number of inclusions with Al-Mg complex deoxidation is larger at an early stage of deoxidation. The critical size of nuclei increases in the order of MgAl_2_O_4_ (0.3–0.4 nm) < Al_2_O_3_ (0.4–0.6 nm), and the nucleation rate increases in the order of Al_2_O_3_ (1100 cm^−3^ s^−1^) < MgAl_2_O_4_ (1200 cm^−3^s^−1^), which is consistent with the experimental results. Moreover, the coarsening rate of MgAl_2_O_4_ inclusions was smaller than Al_2_O_3_ inclusions in both the value of *k_d(cal.)_* from ostwald growth and the value of *k_d(obs.)_* from inclusion size. The effect of Mg addition on coarsening of inclusion was analyzed and their mechanism was discussed based on ostwald ripening theory and Factsage calculation.

## 1. Introduction

Nonmetallic inclusions are frequently reported as the origins of steel performance issues. The control of inclusions is one of the main tasks of steelmakers during the production process to improve the mechanical properties, processing properties, and service life of steel. It is found that the small oxide inclusions with uniform distribution can induce nucleation of intragranular ferrite effectively, which will improve the mechanical properties of steel [1,2]. Due to its strong deoxidization ability and low price, aluminum become the most popular deoxidizer applied in the modern steelmaking process. Nevertheless, the generated Al_2_O_3_ inclusions tend to aggregate and form large clusters that are detrimental to the mechanical properties of steel and easily cause nozzle clogging [3,4]. Therefore, calcium treatment is often applied during refining to modify Al_2_O_3_ inclusions into low-melting calcium aluminates, which is an effective method to prevent the inclusions from clogging the nozzles [5,6,7]. However, it is reported that large size inclusions still can be detected in steel after calcium treatment and these inclusions will be prolonged to tens of microns long after rolling [8]. Hence, a large number of studies were carried out to solve this problem and it was found that Mg treatment has a significant improvement on the steel performances [9,10,11,12]. Lots of inclusions with small size have been observed in the steels with Mg treatment, and it is found that Mg treatment facilitates the increase of inclusion number and the degree of homogeneity in inclusion dispersion [13,14,15,16]. Fu et al. [12] comparatively studied 35CrNi3MoV steel with Al deoxidized and Al-Mg complex deoxidized and found that Mg treatment is conducive to decrease of inclusion size. Kimura et al. [17] observed that MgO and MgAl_2_O_4_ inclusions had a much weaker tendency to coagulate and form clusters via confocal scanning laser microscope. Though lots of studies about Mg treatment have been done, almost all of these studies only revealed phenomenon observed at experiences. The present study explores the nucleation and growth of inclusions.

In the view of controlling size and number of inclusions, the nucleation and growth of inclusions during the deoxidation process in the steel must be considered. Suito and Ohta et al. [18] found that the inclusions are deemed to grow and coarsen by the following steps: the diffusion of reactants to the oxide nuclei, Ostwald ripening, collision, and subsequent coagulation in liquid metal. Kluken [19] and Suzuki et al. [20] concluded that the growth of inclusions in steel should be explained by Ostwald ripening. In spite of extensive studies on formation mechanism and composition analysis [21,22,23,24,25], there are limited studies on nucleation and growth of inclusions containing Mg.

In the current study, the number and size of inclusions were analyzed in Al deoxidized steel and Al-Mg complex deoxidized steel. In addition, the critical nucleation size, nucleation rate, and coarsening rate were calculated based nucleation theory, Ostwald ripening theory, and thermodynamics calculation by FACTSAGE (FACTSAGE (FACTSAGE7.2, Thermfact/CRCT and GTT-Technologies, Montréal and herzogenrath, Canada and Germany) is a thermodynamic calculation software, which is made joint development by McGill University and ecole Polytechniquede Montreal. It combined Facility for the Analysis of Chemical Thermodynamics (FACT-Win) with SOLGASMIX (ChemSage, (ChemSage, GTT-Technologies, herzogenrath, Germany) in 2001 and named as FACTSAGE.). The present study explores the nucleation and growth of inclusions at deoxidized experiments with 0.05%Al and 0.05%Al + 0.03%Mg). In addition, the effect of Mg addition on coarsening of inclusion was analyzed and their mechanism was discussed based on Ostwald ripening theory and Factsage calculation. This study will provide information to understand the relations among characteristics, nucleation, and coarsening of inclusions in Fe-O-Al-Mg melt. The conclusions will be helpful for predicting and controlling size of inclusions.

## 2. Experimental

### Experimental Procedure

In the current study, YT01 pure iron was applied as raw material, and chemical compositions are shown in Table 1. The detailed experimental methods in this work have been described in Wang’s paper [15].

In melting experiments, the Si-Mo heating electric resistance furnace(Braveman Special Testing Furnace CO. LTD., Luoyang, Henan, China) was applied. YT01 pure iron material was enclosed in the Al_2_O_3_ crucible. Meanwhile, the melt was full of Ar gas flowing atmosphere for homogeneity completely, and the melt was held for 30 min after being heated to 1873 K (1600 °C). Then, the Al powder and Ni/Mg alloy packed in iron foil were added immediately. In order to disperse the inclusions uniformly, the melt was stirred by a molybdenum rod for 10 s. The experimental samples were taken by a quartz tube at 120, 600, and 1800 s after deoxidizer addition, followed by quenching in salt water. Before being inserted into the molten steel, the quartz tube was injected with Ar gas to prevent reoxidation of samples.

## 3. Measurement of Inclusions

The main chemical compositions of steel melt, as shown in Table 2, were determined by fusion-infrared absorption and ICP-AES(Inductively Coupled Plasma-Atomic Emission Spectrometry) method. The total oxygen contents in the samples were determined by fusion-infrared absorption, and each sample was measured three times. Total Al, soluble Al, and total Mg contents in steel were analyzed by the ICP-AES method. The dissolved oxygen was calculated by FACTSAGE7.2 based on the chemical composition of melts.

In order to analyze characteristics of inclusions, 169 SEM microphotographs (13 × 13) were taken in each sample. The planar size and number of inclusions were analyzed by Image-Pro Plus software (Image-Pro Plus6.0, Rockville, Media Cybernetics, MD, USA). Due to the limitation of resolution, the small inclusions within 200 nm have not been analyzed.

## 4. Results and Discussion

### 4.1. Composition and Morphologies of Inclusion

SEM-EDS was used for detecting compositions of 50 inclusions in samples at 120, 600, and 1800 s. As shown in Figure 1, the average content of MgO in inclusions at 120, 600, and 1800 s was 19.3%, 19.7%, 18.8%, respectively. According to the Al_2_O_3_-MgO phase diagram [26] at 1873 K (1600 °C), inclusions exist as Al_2_O_3_ + spinel, spinel, and spinel + MgO when MgO is below 16%, in the range 16% to 28% and above 28%, respectively. In the current work, deoxidation product with Al-Mg were MgAl_2_O_4_ inclusions.

The morphologies of typical inclusions of Al_2_O_3_ and MgAl_2_O_4_ as shown at Figure 2. It is obvious that Al_2_O_3_ inclusions are easy to aggregate, while MgAl_2_O_4_ inclusions have a weaker tendency to aggregate. In addition, the size of Al_2_O_3_ inclusions are larger than MgAl_2_O_4_ inclusions.

### 4.2. Characteristics of Inclusions

The two-dimensional size distribution of inclusions as a function of the holding time with Al deoxidation and with Al-Mg complex deoxidation are shown in Figure 3. *N_A_* represents the number density in a certain range of inclusion size and it is obtained by calculating the quotient of the number of inclusions in a sample and the area of sample. Based on references [18,25], the inclusion size distribution tends to be log-normal curves. It is close to log-normal distribution for inclusions with Al deoxidation, but half-normal curves for inclusions with Al-Mg complex deoxidation at 120 s, which indicates that the inclusions were smaller than 0.2-μm generates with Al-Mg complex deoxidation at 120 s. Meanwhile, the small inclusions with Al-Mg complex deoxidation generate indicates that the initial size of inclusions were small and the critical size of nuclei was small. After deoxidation for 120 s, the number density with Al-Mg complex deoxidation at the range from 0.2–0.6 μm is approximately equal to the number density with Al deoxidation. Therefore, there is a slight difference in the number of inclusions between Al deoxidation and Al-Mg complex deoxidation, which indicates that the nucleation rate of inclusions with Al-Mg complex deoxidation is close to the nucleation rate of inclusions with Al deoxidation. However, compared with inclusions deoxidated by Al, the number density of large size inclusions is lower with Al-Mg complex deoxidation at 120, 600, and 1800 s, respectively. It shows that inclusions with Al deoxidation are easy to coarse. In addition, it is found that the proportion of large size inclusions increases with the holding time as a result of growth or collision. There were inclusions with 4.6 μm with Al deoxidation at holding time of 120 s, it suggests that Al_2_O_3_ inclusions aggregated rapidly.

Figure 4 shows that the average size of inclusions increases generally, and the number of inclusions decreases with holding time elapsed. After deoxidation for 120 s, the average size of inclusions with Al deoxidation is 1.3 times as many as inclusions with Al-Mg complex deoxidation, which indicates that the critical size of nuclei of inclusions with Al-Mg complex deoxidation is smaller than inclusions with Al deoxidation. Though the number of inclusions deoxidated by Al-Mg is lower than those deoxidated by Al, as shown in Figure 4, the inclusion size distribution tends to be half-normal curves for inclusions with Al-Mg complex deoxidation, as shown in Figure 3b. Therefore, the actual number of inclusions with Al-Mg complex deoxidation is approximately 2 times as many as the observed inclusions shown in Figure 4, which indicates that the number of inclusions deoxidated by Al-Mg is larger than those by Al. In addition, at the process from 600 s to 1800 s, the average size increases by 0.2 with Al-Mg complex deoxidation and 0.5 with Al deoxidation, respectively, which indicates that the coarsening rate of inclusions with Al deoxidation is larger than inclusions with Al-Mg complex deoxidation. However, at the process from 120 s to 1800 s, the rate of increase for average size with Al deoxidation is larger than inclusions with Al-Mg complex deoxidation as a result of flotation of large size Al_2_O_3_ inclusions.

It is concluded that small inclusions were more easily found in the steels with Al-Mg complex deoxidation. The size distribution of inclusions shows that the inclusions smaller than 0.2 μm generate with Al-Mg complex deoxidation at 120 s. The average size of inclusions increases generally, and the number of inclusions decreases with holding time elapsed. At early stage of deoxidation, the number of inclusions deoxidated by Al-Mg is larger than by Al, which indicates that the nucleation rate of inclusions with Al-Mg complex deoxidation is larger than inclusions with Al deoxidation, and the smaller average size of inclusions with Al-Mg complex deoxidation indicates that the critical size of nuclei (the critical size that transfers unstable embryos into stable nuclei) is smaller with Al-Mg complex deoxidation. In addition, the larger change of inclusion size with Al deoxidation indicates that the coarsening rate of inclusions with Al deoxidation is larger. The lower number density of large size inclusions with Al-Mg complex deoxidation at 120, 600, and 1800 s, respectively, indicates that inclusions with Al deoxidation are easy to coarse.

## 5. Calculation

### 5.1. Calculated Nucleation Rate and Critical Size of Nuclei

In order to study the contents of Al, Mg, and O on the nucleation rate, *I* (cm^−3^s^−1^) was estimated as the following relationship based on the classical nucleation theory [18]:(1)$$\ln I=\frac{16{\pi \gamma }_{\mathrm{SL}}^{3}V_{O}^{2}}{3k_{B}R^{2}T^{3}}\left(\frac{1}{{\left({\mathrm{lnS}}_{O}^{*}\right)}^{2}}-\frac{1}{{\left({\mathrm{lnS}}_{O}\right)}^{2}}\right)$$
where *V_O_* is the molar volume of oxide (m^3^/mol); $S_{O}^{*}$ is the critical supersaturation degree, which is value of *S_O_* at *I* = 1 cm^−3^ s^−1^; *k_B_* is the Boltzman constant (1.38 × 10^−23^ J/K); *R* is the gas constant (8.314 J·mol^−1^·K^−1^); and *T* is the absolute temperature (K).

Based on classic homogeneous nucleation theory, the critical size of nucleation *r*_*C*_ [18] is given by
(2)$$r_{C}=-\frac{2\gamma_{SL}}{\Delta G_{V}}=\frac{2r_{SL}V_{O}}{{\mathrm{RTlnS}}_{O}}.$$

$\gamma_{SL}$ is the interfacial energy between oxide and liquid steel (J/m^2^), and it can be expressed by Young’s Equation:(3)$$\gamma_{SL}=\gamma_{SV}-\gamma_{LV}\cos\theta ,$$
(4)$$\gamma_{LV}=\gamma_{O}-\sum \gamma_{Fe}^{i},$$
(5)$$\gamma_{O}=2.858-0.000591T,$$
where *γ*_*S**V*_ is the surface energy of solid inclusion, *γ*_*L**V*_ is the surface energy of the liquid steel, *θ* is the contact angle of liquid steel on solid oxide, *γ*_*O*_ energy of pure liquid iron, and $\gamma_{Fe}^{i}$ is the effect of steel composition on the surface energy of the liquid steel. Substituting the relevant data in Table 1 and Table 3, the following relationship is derived:(6)$$\gamma_{LV}=1.75-0.279\ln\left(1+140\times a_{O}\right).$$

The calculated critical size of nuclei and nucleation rate for oxide inclusions at 1873 K (1600 °C) are shown in Figure 5. The critical size of nuclei is obtained by substituting Equation (3) and relevant data in Table 4 to Equation (2), and then the value of *S_O_* is known when the *r_C_* is a certain value. According to the relationship between *S_O_* and *K_MO_* (*S_O_* = *K_MO_*/*K_equilibrium.(O)_*), it is easy to obtain the relationship between *a_O_* and *K_MO_,* as shown in Figure 5. *K_eq.(O)_* is the equilibrium constant per mole oxide, as shown in Table 5, and the *K_MO_* is solubility product per mole oxide (*K_MO_* = *a_M_1/Xa_O_*, in this work, *K_Al2O3_* = *a_Al_2/3a_O_*, *K_MgAl2O4_* = *a_Mg_1/4a_Al_1/2a_O_*, *K_MgO_* = *a_Mg_a_O_*). The method to obtain the nucleation rate is similar to the critical size of nuclei by substituting relevant data to Equation (1). In Figure 5, the red dotted lines are the critical size of nuclei, the blue solid lines are the nucleation rate and the black spherical marks are the *r_C_* and *lnI* value based on the compositions of experimental melts. The calculated results show that there is little difference in the critical size of nuclei for Al_2_O_3_, MgAl_2_O_4_, and MgO with size from 0.1 to 2 nm. It indicates that the nucleation rate is strongly dependent on the activity of oxygen when *a*o exceeds a certain value and the nucleation rate for Al_2_O_3_, and MgAl_2_O_4_ increases with the increasing *a*o. The critical size of nuclei for Al_2_O_3_ and MgAl_2_O_4_ decreases with the increasing *a*o and *K_MO_*, and MgO increases first and then decreases. Based on the compositions of melt, the calculated results show that the critical size of nuclei at holding time of 120 s increases in the order of MgAl_2_O_4_ (0.3–0.4 nm) < Al_2_O_3_ (0.4–0.6 nm) and the nucleation rate increases in the order of Al_2_O_3_ (1100 cm^−3^ s^−1^) < MgAl_2_O_4_ (1200 cm^−3^ s^−1^). At the early stage of deoxidation, the calculated results show that the size of Al_2_O_3_ inclusions is larger than MgAl_2_O_4_ inclusions, and the nucleation number of MgAl_2_O_4_ inclusions is larger than Al_2_O_3_ inclusions.

The compositions of melt as shown in Table 2 and the activities of O, Al, and Mg are obtained by substituting compositions of melt and relevant thermodynamic data in Table 6 to Equations (7) and (8) [36]. The estimated composition of steels was shown in Table 7.
(7)$$\;a_{i}=f_{i}\left[mass\;\%i\right],$$
(8)$$\log f_{i}=\sum e_{i}^{j}\left[mass\;\%i\right],$$
where *a_i_*, *f_i_*, and [*mass* %*i*] are the 1 mass% activity, 1 mass% activity coefficient, and the concentration of i in mass fraction, respectively. $e_{i}^{j}$ is the first-order interaction coefficient. The calculated results show that the critical size of nuclei of inclusions with Al deoxidation is larger than inclusions with Al-Mg complex deoxidation, and the nucleation rate of inclusions with Al deoxidation is smaller than inclusions with Al-Mg complex deoxidation. It is concluded that the calculated results based on homogeneous nucleation theory are consistent with the experimental results.

### 5.2. Ostwald Growth

Based on the research of Ohta [27], the inclusions growth by Ostwald ripening can be expressed by
(9)$${\bar{r}}^{3}-{\bar{r_{0}}}^{3}=k_{d}\times \alpha \times t,$$
(10)$$\;k_{d\left(O\right)}=\frac{2\gamma_{SL}D_{O}V_{O}C_{O}}{RT\left(C_{P}-C_{O}\right)},\;$$
where $\bar{r}$ and $\bar{r_{0}}$ is the mean inclusions radius at time t(m) and that at the start of Ostwald growth respectively, *k_d_* is coarsening rate(μm3·s^−1^), α is the coarsening coefficient (α = 4/9 in LSW theory), *D_O_* is the diffusion constant of oxygen (2.91 × 10^−9^ m^2^/s), *C_O_* is the dissolved oxygen concentration expressed by weight per unit volume (kg/m^3^), and *C_P_* is the oxygen concentration in oxide expressed by weight per unit volume (kg/m^3^).

*k_d(cal.)_*_/_*k_d(calculation)_* from Ostwald growth and *k_d(obs.)_*_/_*k_d(observation)_* from inclusion size was obtained by considering the oxygen diffusion, as shown in Figure 6. *k_d(obs.)_* can be expressed by Equation (9) and *k_d(cal.)_* can be expressed by Equation (10). Figure 6 shows that the coarsening rate of MgAl_2_O_4_ inclusions was smaller than Al_2_O_3_ inclusions whether *k_d(cal.)_* from Ostwald growth or *k_d(obs.)_* from inclusion size. Though there are half points lying at line *k_d(cal.)_* = *k_d(obs.)_*, the difference between *k_d(cal.)_* and *k_d(obs.)_* is controlled within an order of magnitude. In addition, the experimental results show that the number of inclusions decreases sharply from 600 s to 1800 s as a result of floatation of large number of inclusions, which affects the calculated results.

It is necessary to consider the diffusion of deoxidation metal element duo to the same order of magnitude between magnesium content and oxygen content. The effect of Mg addition on the Ostwald growth of inclusions with [Al] of 0.04% and 0.03% was obtained by considering the oxygen diffusion and magnesium diffusion, as shown in Figure 7. The coarsening rate *k_d(Mg/Al)_* can be expressed by Equation (11), in which the notations are similar to Equation (10). *D_Mg/Al_* is assumed to be equal to the *D_O_* because the solute diffusivities in liquid Fe are considered to be the same order of magnitude [22].
(11)$$\;k_{d\left(M\right)}=\frac{2\gamma_{SL}D_{M}V_{M}C_{M}}{RT\left(C_{P\left(M\right)}-C_{M}\right)}.$$

Figure 7 is obtained based on Equations (10) and (11). FACTSAGE7.2 software is used in current study. Equilibrium compositions of melt with Mg addition are estimated by FACTSAGE7.2 with the FToxid and FTmisc databases (based on the compositions of raw materials). “Equilib” module is used, and pure solids Fe-liq in solution phases are selected as products. Calculated temperature and pressure are set as 1600 °C and 1 atm, respectively. Then, the content of O, Al, Mg, Al_2_O_3_, Spinel, and MgO are obtained by FACTSAGE7.2. The coarsening rates of Al_2_O_3_, Spinel, and MgO are the smallest values found by comparing *k_d(O)_* and *k_d(M)_*. The eventual value of *k_d_*, as shown in Figure 7, is obtained by calculated the sum of products of the percentage of molar mass of Al_2_O_3_, Spinel, and MgO by the coarsening rates of Al_2_O_3_, Spinel, and MgO, respectively. The coarsening rate of inclusions decreases when the amount of added Mg under 0.0005% decreases at the range of 0.0005%–0.0045%, increases at the range of 0.0045%–0.0085%, and keeps steady beyond 0.0085%. During the four stages shown in Figure 7, the products are Al_2_O_3_ determined by oxygen diffusion; Al_2_O_3_+ Spinel determined by Mg and oxygen diffusion; Spinel determined by Mg diffusion; MgO+ Spinel determined by Mg and oxygen diffusion, respectively. (An explanation must be made between Figure 5 and Figure 7. The parameters at tables are only used for nucleation calculation as shown in Figure 5, while the parameters in the FToxid and FTmisc databases are only used for coarsening rate calculation preliminarily as shown in Figure 7. At the section of nucleation calculation, the parameters at tables are derived from Thermodynamic Data for Steelmaking [36], in which the data are new and verified. At the section of coarsening rate calculation, it is difficult to calculate via classical thermodynamics but Factsage7.2 at multicomponent equilibrium. The data in the FToxid and FTmisc databases may not be accurate, but the tendency of calculated results is consistent. In fact, it is worth replacing the data in the FToxid and FTmisc databases with updated data at calculation research, and the Compound module at Factsage7.2 must be used. In future work, the relevant research will be explored.)

## 6. Conclusions

In the current study, experiments and nucleation calculation were performed to investigate the effect of Mg treatment on nucleation and Ostwald growth in Fe-O-Al-Mg melt. Based on the experimental results and nucleation analysis, the following conclusions were obtained. (1) It is concluded that small inclusions were more easily found in the steels with Al-Mg complex deoxidation. The size distribution of inclusions shows that the inclusions smaller than 0.2 μm generate with Al-Mg complex deoxidation at 120 s. The average size of inclusions increases generally, and the number of inclusions decreases with holding time elapsed. At early stage of deoxidation, the number of inclusions deoxidated by Al-Mg is larger than those by Al, which indicates that the nucleation rate of inclusions with Al-Mg complex deoxidation is larger than inclusions with Al deoxidation, and the smaller average size of inclusions with Al-Mg complex deoxidation indicates that the critical size of nuclei is smaller with Al-Mg complex deoxidation. In addition, the larger change of inclusion size with Al deoxidation indicates that the coarsening rate of inclusions with Al deoxidation is larger. The lower number density of large size inclusions with Al-Mg complex deoxidation at 120, 600, 1800 s respectively indicates that inclusions with Al deoxidation are easy to coarse. (2) The calculated results show that the critical size of nuclei at a holding time of 120 s increases in the order of MgAl_2_O_4_ (0.3–0.4 nm) < Al_2_O_3_ (0.4–0.6 nm). The nucleation rate increases in the order of Al_2_O_3_ (1100 cm^−3^s^−1^) < MgAl_2_O_4_ (1200 cm^−3^s^−1^). It is concluded that the calculated results based on homogeneous nucleation theory are consistent with the experimental results. (3) The coarsening rate of MgAl_2_O_4_ inclusions was smaller than Al_2_O_3_ inclusions whether using *k_d(cal.)_* from Ostwald growth or *k_d(obs.)_* from inclusion size. Based on Ostwald ripening theory and Factsage calculation, the effect of Mg addition on coarsening of inclusion was analyzed and their mechanism was discussed. The coarsening rate of inclusions decreases when the amount of added Mg is under 0.0005%, decreases at the range of 0.0005%–0.0045%, increases at the range of 0.0045%–0.0085%, and keeps steady beyond 0.0085%. During four stages, the products are Al_2_O_3_ determined by oxygen diffusion; Al_2_O_3_+ Spinel determined by Mg and oxygen diffusion; Spinel determined by Mg diffusion; and MgO+ Spinel determined by Mg and oxygen diffusion, respectively.

## Figures and Tables

**Figure 1 materials-13-03355-f001:**
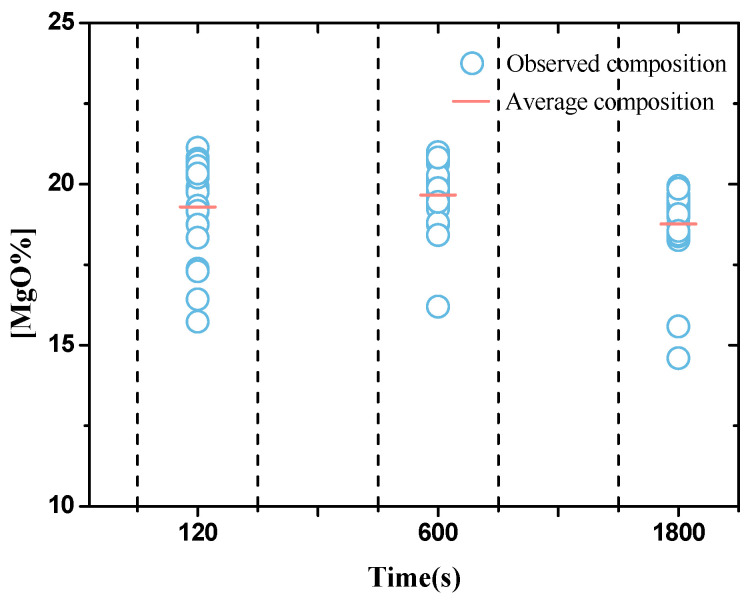
Composition of inclusions in steel with holding time.

**Figure 2 materials-13-03355-f002:**
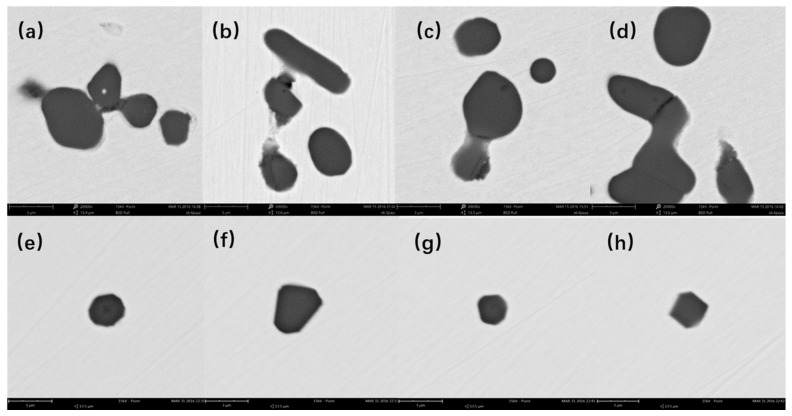
Morphologies of typical inclusions in samples by SEM: Al_2_O_3_ (**a**–**d**); MgAl_2_O_4_ (**e**–**h**).

**Figure 3 materials-13-03355-f003:**
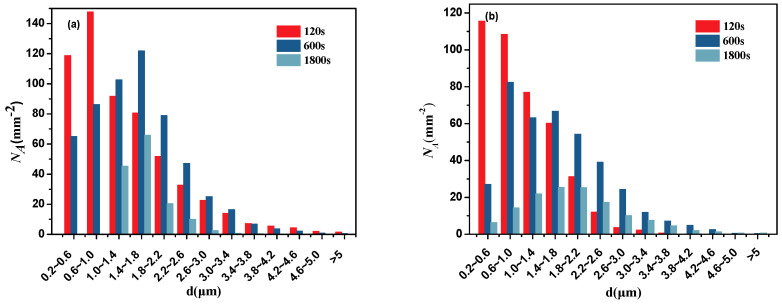
Two-dimensional size distribution of inclusions as a function of holding time. (**a**) Al deoxidation, (**b**) Al-Mg complex deoxidation.

**Figure 4 materials-13-03355-f004:**
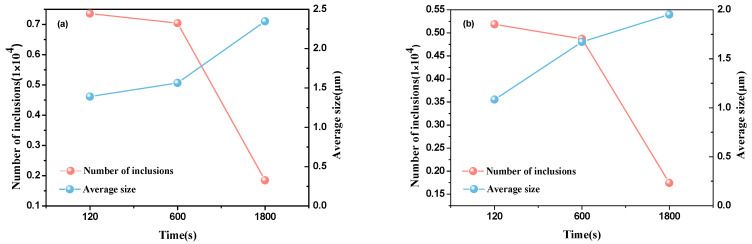
The number and the average size of inclusions as function of holding time. (**a**) Al deoxidation, (**b**) Al-Mg complex deoxidation.

**Figure 5 materials-13-03355-f005:**
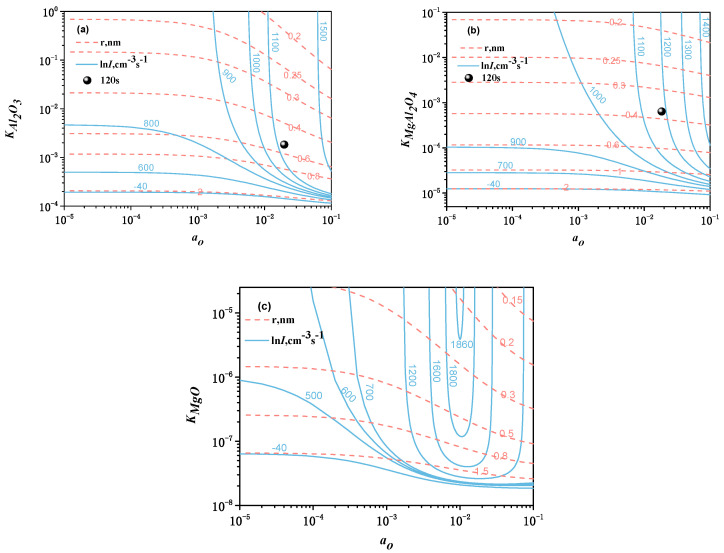
Calculated critical size of nuclei and nucleation rate for oxide inclusions at 1873 K: (**a**)Al_2_O_3_, (**b**) MgAl_2_O_4_, (**c**) MgO.

**Figure 6 materials-13-03355-f006:**
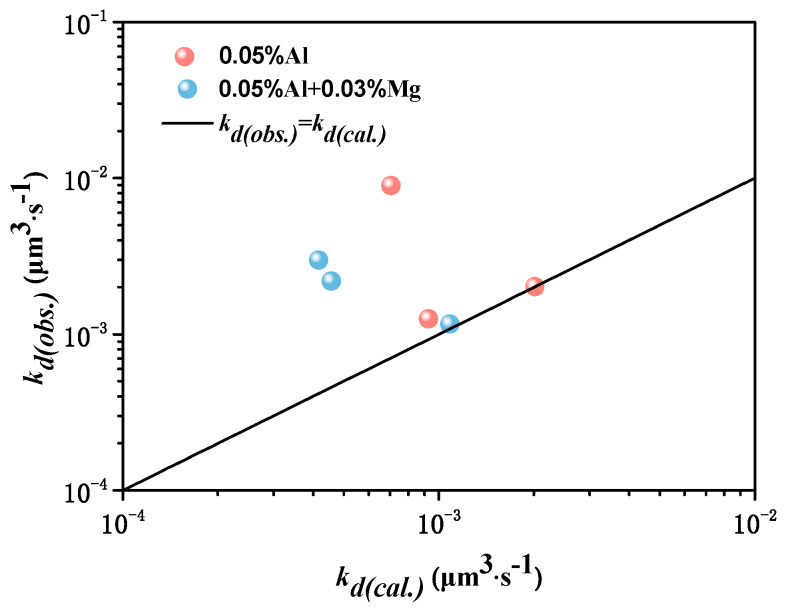
Comparison of k_d(cal.)_ from Ostwald growth and k_d(obs.)_ from inclusion size.

**Figure 7 materials-13-03355-f007:**
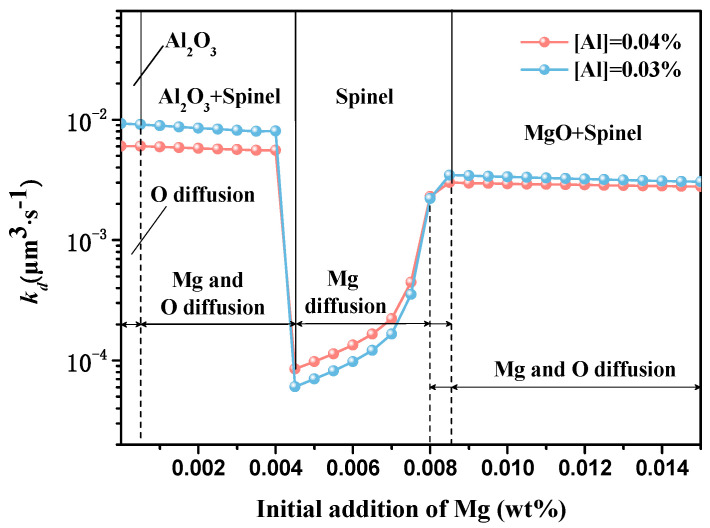
Effect of Mg addition on coarsening rate caused by Ostwald ripening k_d_ of inclusions.

**Table 1 materials-13-03355-t001:** Chemical compositions of mother steel (weight percent).

Element	C	Si	Mn	P	S	Cr	Al	Cu	Ni	Ti	N
Content	0.0016	0.0033	0.01	0.0053	0.0017	0.0107	0.003	0.0037	0.0038	0.001	0.0020

**Table 2 materials-13-03355-t002:** Chemical composition of melt.

Deoxidizer	Holding Time	[O]		[Al]		[Mg]	
		Total	Sol.	Sol.	Insol.	Sol.	Insol.
		(Mass ppm)	(Mass ppm)	(Mass ppm)	(Mass ppm)	(Mass ppm)	(Mass ppm)
0.05%Al	120 s	218	7.32	310	94	-	-
	600 s	90.9–92.8	3.6	-	-	-	-
	1800 s	14.7–117	3.38	280	79	-	-
0.05%Al+0.03%Mg	120 s	208	4.08	390	85	8–19	-
	600 s	162	3.44	-	-	-	-
	1800 s	63	2.93	350	23	5	-

ppm means parts per million. Sol. for [Al]/[Mg] means soluble [Al]/[Mg] in acid. Insol. for [Al]/[Mg] means insoluble [Al]/[Mg] in acid.

**Table 3 materials-13-03355-t003:** Effect of steel composition on surface energy of liquid steel (J/m^2^).

**Element**	**C**	**Si**	**Mn**	**P**	**N**	**Al**
$\gamma_{\mathrm{Fe}}^{i}$	0.065[C pct] [27]	0.026[Si pct] [28]	0.05[Mn pct] [28]	0.025[P pct] [28]	5.585[N pct] [28]	0.037[Al pct] [28]
**Element**	**Cr**	**Cu**	**Ni**	**S**	**O**	
$\gamma_{\mathrm{Fe}}^{i}$	0.008[Cr pct] [28]	0.026[Cu pct] [28]	0.002[Ni pct] [28]	0.2ln(1+330[pct S]) [29]	0.279 ln(1+140[$a_{O}$]) [30]	

pct is abbreviation for per cent.

**Table 4 materials-13-03355-t004:** Parameters used in the calculation of critical size of nuclei and nucleation rate.

Oxide	Θ (deg)	γ_SV_ (J/m^2^)	V_O_ (m^3^/mol)
Al_2_O_3_	132 − 6.3ln(1 + 400[pctO])0.63ln(1 + 640[pctS]) [29]	1.128 − 0.0001T [30]	8.6 × 10^−6^
MgO	117 − 7.4ln(1 + 720*a*_*O*_) (−15 < log*a*_*O*_ < 9) [31]	0.86 [32]	11 × 10^−6^
MgAl_2_O_4_	105 [33]	2.270 − 0.0006T [34,35]	9.3 × 10^−6^

**Table 5 materials-13-03355-t005:** Equilibrium constants used in this study.

Equation	logK_eq_
Al_2_O_3_(s) = 2[Al] + 3[O]	−12.57 = (−45300/T + 11.62) [36]
MgO(s) = [Mg] + [O]	−7.86 = (−38059/T + 12.45) [36]
MgAl_2_O_4_(s) = [Mg] + 2[Al] + 4[O]	−21.28 = (−84339/T + 23.75) [36]

**Table 6 materials-13-03355-t006:** Interaction coefficients of O, Al, and Mg at 1873 K (1600 °C).

$e_{i}^{j}(\to j)$	C	Si	Mn	P	S	Cr	Cu	Ni	Ti	N	O	Al	Mg
O	−0.42	−0.066	−0.021	0.07	−0.13	−0.055	−0.013	0.006	−0.34	−0.14	−0.17	−1.17	−1.98
Al	0.091	0.056	−0.004	0.033	0.035	0.012	−0.013	−0.017	-	0.015	−1.98	0.043	−0.13
Mg	−0.31	−0.088	-	-	-	0.047		−0.012	−0.64	-	−3	−0.12	-

**Table 7 materials-13-03355-t007:** Estimated composition of steels (weight percent).

Holding Time	[O]	[Al]	[Mg]	*a* _O_	*a* _Al_	*a* _Mg_
120 s	0.0218	0.0310	0	0.019783	0.028190	0
1800 s	0.0046	0.0280	0	0.004237	0.027531	0
120 s	0.0208	0.0390	0.0008−0.0019	0.018367	0.035641	0.001154
1800 s	0.0063	0.0350	0.0005	0.005677	0.034167	0.000473

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
