# Peer review of "Effect of Mg Treatment on the Nucleation and Ostwald Growth of Inclusions in Fe-O-Al-Mg Melt"

_materials, 2020, doi:10.3390/ma13153355_

Round 1

Reviewer 1 Report

The results obtained by the authors certainly have both theoretical and practical value. Therefore, they can be published.
However, there are a number of comments to the work.
Design Notes
1. The tables are in the following order: 1, 2, 5, 3, 4, 6, 7. Why is this order?
2. Quoting literary sources. After 26, the citation sequence is lost.
3. The link to table 6 is for some reason framed as VI. What for?

Methodological notes
1. The main methodological problem is that in fact, within the framework of one work, different sets of thermodynamic parameters of the dissolution of elements in liquid iron were used. One set is explicitly presented in Tables 4 and 6. Another set is implicitly contained in the FT-misc database. This is probably an unacceptable contradiction.They (the sets) are probably different and maybe (we don't know) very different. It was possible to use only FT-misc and extract the data necessary for other calculations from it (calculate the activity of the components at different concentrations, from which you can get both Wagner parameters and equilibrium constants if they are very needed).

2. It is not clear from what considerations the values ​​in tables 4 and 6 are selected. The literature contains a lot of such data and sometimes they differ by tens or hundreds of times. The choice made by the authors needs justification.

Reviewer 2 Report

In the article experimental and calculated state of inclusions in tested steel after Al and Al-Mg complex deoxidation was analysed.

In my opinion the scope of work corresponds to the scope of the 'Materials'. 

General and particular remarks.

  • Introduction. Is it optimal state of inclusions (size, localisation, distribution) ? How it was earlier analysed and experimentally verified ?
  • Introduction. In my opinion novelty of the work should be more precise emphasized.
  • Lines 52-53. 'FACTSAGE' should be explained (any reference ?).
  • Line 57. What means "procedure" ?
  • Lines 71-72. Sentence style. Fusion-infrared absorption and ICP-AES method are not shown in Table 2.
  • Table 2. Abbreviations: ppm, Sol., Insol. should be explained.
  • Lines 83-84. Example SEM images used for EDS analysis should be added.
  • Line 93, Fig.2. Unit of Na is unclear (mm-2). It should be explained.
  • Lines 115-116. Probably, there should be another number of the Figure in the sentence: '... the inclusion size distribution tends to be half-normal curves for inclusions with Al-Mg complex deoxidation as shown in Fig.2'.
  • Lines 117-119. 'Therefore, the actual number of inclusions with Al-Mg complex deoxidation is approximately 2 times as many as the observed inclusions as shown in Fig.3, which indicates that the number of inclusions deoxidated by Al-Mg is larger than by Al'. Could you prove it ? Figure 3 shows the similar number and sizes of inclusions for Al and Al-Mg. Even if the distribution of inclusion size is another (Fig.2).
  • Line 132. What is the critical size of nuclei ?

Round 2

Reviewer 1 Report

I am not completely satisfied with the explanations of the authors on points 4 and 5.

In particular, on point 4, in my opinion, a reservation should be made in the text of the article that different sets of parameters are used for different parts of the calculations. And there is some problem here. But you are counting on that this did not significantly affect the main results of the work. In my opinion, such a reservation would be useful to the readers.
